# The Multifaceted Role of Flavonoids in Cancer Therapy: Leveraging Autophagy with a Double-Edged Sword

**DOI:** 10.3390/antiox10071138

**Published:** 2021-07-19

**Authors:** Zhe Zhang, Jiayan Shi, Edouard C. Nice, Canhua Huang, Zheng Shi

**Affiliations:** 1Clinical Medical College & Affiliated Hospital of Chengdu University, Chengdu University, Chengdu 610106, China; scuzz@stu.scu.edu.cn (Z.Z.); shijiayan@stu.scu.edu.cn (J.S.); 2Collaborative Innovation Center for Biotherapy, Cancer Center and State Key Laboratory of Biotherapy, West China Hospital, and West China School of Basic Medical Sciences & Forensic Medicine, Sichuan University, Chengdu 610041, China; 3Department of Biochemistry and Molecular Biology, Monash University, Clayton, VIC 3800, Australia; ed.nice@monash.edu

**Keywords:** flavonoids, autophagy, cancer prevention, cancer therapy, nanotechnology

## Abstract

Flavonoids are considered as pleiotropic, safe, and readily obtainable molecules. A large number of recent studies have proposed that flavonoids have potential in the treatment of tumors by the modulation of autophagy. In many cases, flavonoids suppress cancer by stimulating excessive autophagy or impairing autophagy flux especially in apoptosis-resistant cancer cells. However, the anti-cancer activity of flavonoids may be attenuated due to the simultaneous induction of protective autophagy. Notably, flavonoids-triggered protective autophagy is becoming a trend for preventing cancer in the clinical setting or for protecting patients from conventional therapeutic side effects in normal tissues. In this review, focusing on the underlying autophagic mechanisms of flavonoids, we hope to provide a new perspective for clinical application of flavonoids in cancer therapy. In addition, we highlight new research ideas for the development of new dosage forms of flavonoids to improve their various pharmacological effects, establishing flavonoids as ideal candidates for cancer prevention and therapy in the clinic.

## 1. Introduction

Natural products are valuable sources for the discovery of drugs for which researchers can isolate new active agents that might serve as leads or scaffolds for the construction of novel efficacious drugs with enhanced biological activity [1,2,3]. Given the remarkable chemical diversity of natural products, they have played a powerful role in developing therapeutic strategies either in their naturally occurring or synthetically modified forms [4]. Such efforts have identified valuable natural products such as paclitaxel and vincristine from plants, and have assessed their properties and explored their mechanisms of action [5,6,7]. Natural products have exhibited good anti-cancer activity with minimal side effects. Therefore, applying natural products in clinical therapy is an effective, inexpensive, and accessible therapeutic approach [8,9]. Additionally, as many conventional chemotherapeutic drugs have failed to achieve the desired result due to chemoresistance and cytotoxicity, novel multi-targeted therapeutic strategies using natural products have become an important strategy [10,11]. Many natural products such as alkaloids, terpenoids, flavonoids, quinones, and steroids are suitable to gain prominence as new anti-cancer drugs for clinical treatment because of their unique chemical structures and pleiotropic activities [12,13].

As a class of natural polyphenolic metabolites, flavonoids are an important group of natural products that are widely present in a variety of plants, especially edible fruits, vegetables, [14] and plant-derived beverages such as green tea, wine, and cocoa-based products [15]. The flavonoids are mainly derived from benzo-γ-pyrone and the basic structure of flavonoids is the flavan skeleton that consists of two aromatic rings (A ring and B ring) and a heterocyclic pyran ring (C ring) [16,17]. Based on the chemical structure, the level of oxidation and pattern of substitution of the C ring on which the B ring is attached, flavonoids can be subdivided into six main subgroups: flavones, flavonols, flavanones, isoflavonoids, anthocyanins, and chalcones [18] (Figure 1). Flavonoids are phenolic compounds that are easily oxidized into quinones; thus, they are capable of protecting ascorbic acid and the unsaturated fatty acids in membranes from oxidation [16]. This process may be accompanied by a ring opening that can easily be conducted under ultraviolet light, especially if heavy metal ions are also present [16]. Recently, flavonoids have been widely reported to exhibit beneficial effects on various diseases including immune diseases, cardiovascular disease, and cancer [19,20,21]. In the context of cancer, a number of preclinical studies have provided a comprehensive perspective on flavonoids that can help to optimize their action in disease prevention and therapy [22]. Notably, these potential applications in the clinic of flavonoids are largely dependent on modulation of autophagy [23]. For example, flavonoids can inhibit cell growth and trigger cell death in various kinds of cancer by inducing excessive and sustained autophagy or impairing autophagy flux, suggesting promising effectiveness for the treatment of patients with apoptosis-resistant cancer [24,25]. By contrast, the anti-cancer activity of flavonoids may be unexpectedly compromised by the induction of protective autophagy, indicating that autophagy inhibition could act as a potential strategy in optimizing cancer therapeutics [26]. The clinical application of flavonoids has perceived advantages in terms of preventing cancer, especially for tumors caused by autophagy defects [15]. Furthermore, flavonoids-triggered protective autophagy is an actionable target for protecting against chemotherapy or radiotherapy-induced undesired side effects in normal tissues [27,28]. Accordingly, understanding the relationship of flavonoids and autophagy in depth may provide a new perspective for clinical applications of flavonoids.

In this review, we mainly focus on the application of flavonoids as potential preventive and therapeutic agents in cancer, highlight the underlying molecular mechanisms, and discuss potential candidates for the proposed administration to tumor patients based on modulating the dual role of autophagy. We also overview the novel approaches that can contribute to the increased bioavailability and selectivity of flavonoids compounds, and discuss the major challenges encountered to date in the application of flavonoids for medication. Our analysis has been based on the use of several databases (PubMed, ClinicalTrials, and Web of Science) including the available data up to June 2021 using the following keywords: flavonoids, autophagy, cancer prevention, and cancer therapy. Studies were selected after applying the following inclusion and exclusion criteria. The inclusion criteria were: (1) papers published in the English language and within the past 20 years; (2) articles with a high citation rate; and (3) studies with broad application prospects. The exclusion criteria were: (1) articles associated with a low credibility rating; (2) papers published in languages other than English; and (3) papers not available as a full-text format.

## 2. Flavonoids as Therapeutic Agents

Excessive and sustained autophagy induced by flavonoids has been exploited as a potential strategy for cancer therapy especially in apoptosis-defective tumor cells. However, the anti-cancer activity of flavonoids may be impaired due to cytoprotective autophagy, suggesting that combining autophagy inhibitors with flavonoids may optimize the clinical effect on tumors. Here, we summarize flavonoid compounds used for cancer therapy and the related mechanisms involved in autophagy to distinguish between agents for suitable administration based on an understanding of the modulation of autophagy (Figure 2).

### 2.1. Inducing Autophagic Cell Death by Flavonoids

The administration of pro-apoptotic strategies in clinical cancer therapy will inevitably lead to the emergence of apoptosis-resistant cells that represents a major obstacle to successful clinical development [29]. Thus, it is imperative to develop novel therapeutic strategies that induce tumor cell death by non-apoptotic mechanisms [30]. Notably, autophagy-mediated cell death can be induced in apoptosis-resistant cells particularly in the absence of the pro-apoptotic proteins Bax and Bak [31,32]. Therefore, it is possible for autophagy to overcome resistance to apoptosis in cancer, allowing the cellular stress response to take the path of autophagic cell death [31,33]. Many flavonoids display anticancer activities by inducing excessive autophagy or impairing autophagy flux by mediating different signaling pathways such as PI3K/Akt/mTOR, AMPK, MAPK, Beclin-1, and Wnt/β-catenin [24]. As resistance to chemotherapeutic agents predominantly occurs through defects in the apoptotic signaling pathway, the identification of flavonoids that can stimulate autophagic cell death may become useful for cancer patients undergoing ineffective treatment due to apoptotic resistance [34]. In the following section, kaempferol and apigenin were taken as two typical examples to illustrate that flavonoids can induce the autophagic death of a variety of cancer cells, thus providing theoretical support for the use of flavonoids to overcome apoptosis-resistant tumors.

#### 2.1.1. Kaempferol

Kaempferol (3,5,7-trihydroxy-2-(4-hydroxyphenyl)-4H-1-benzopyran-4-one) is one of the most common dietary flavonols and is mainly derived from the rhizomes of *Kaempferia galanga L.* that are used in traditional medicine in Asia [35,36,37]. Preclinical studies demonstrated noticeable anti-cancer effects, suggesting a promising application role in the clinic [38,39]. For example, kaempferol induced cell death by activation of IRE1-JNK-CHOP signaling-mediated ER stress in gastric cancer cells, eliciting epigenetic change and autophagic cell death by a HDAC/G9a axis-dependent way [40]. AMPK was reported as another upstream mediator mediating kaempferol-induced autophagic cell death in hepatocellular carcinoma cells. Specifically, kaempferol enhanced AMPKα1 stabilization by regulating its ubiquitin ligase and melanoma antigen 6, resulting in AMPK-induced Ulk1 phosphorylation, mTOR complex 1 inhibition, and eventually persistent autophagy [41]. In agreement, a previous study reported kaempferol-induced autophagic cell death in human hepatic cancer cells that was partially attributed to the activation of AMPK signaling pathways [42]. In addition, Han et al. [43] suggested the application of kaempferol for lung cancer whereby upregulation of miR-340 by kaempferol treatment lead to the inhibition of the PTEN/PI3K/AKT pathway, contributing to excessive autophagy. Taken together, these studies demonstrate that kaempferol displays considerable promise for cancer therapy by modulating autophagy, although further clinical application is required to obtain a more complete understanding of the anti-cancer mechanisms involved.

#### 2.1.2. Apigenin

Apigenin (4′,5,7-trihydroxyflavone) is a predominant monomeric flavonoid that belongs to the flavone sub-class [44]. Widely distributed in edible plants, especially in celery, apigenin can be easily ingested from a daily diet [45]. Importantly, apigenin has remarkable anticancer activity with low toxicity in normal cells [46]. From studies on the anticancer mechanisms of apigenin, researchers found that the administration of apigenin correlated with a high level of intracellular ROS and activated autophagy [47]. Kang et al. [48] reported that the generation of ROS induced by apigenin subsequently augmented autophagic flux. Recent studies clearly demonstrated that apigenin killed multiple myeloma cells by triggering autophagy to a certain extent [49,50]. In agreement with these studies, Kim et al. [51] reported that apigenin treatment induced autophagy resulted in obvious inhibition of gastric cancer. Mechanistically, inhibition of PI3K/Akt/mTOR cascades, frequently related to excessive autophagy activation, resulted from apigenin exposure [51]. Furthermore, apigenin exhibited significant suppression of the growth of cisplatin-resistant colorectal cancer cells that was shown to be associated with PI3K/Akt/mTOR signaling-mediated autophagy [52]. Apigenin-induced autophagic cell death was also confirmed in human papillary thyroid carcinoma BCPAP cells [53]. Although there is a lack of relevant antitumor clinical trials, apigenin is being used as a co- therapeutic agent in the treatment of COVID-19 patients, potentially associated with the regulation of oxidative stress and autophagy [54]. This may provide some guidance for antitumor clinical studies with apigenin (NCT04873349). We expect that further preclinical and clinical trials will support the translation of apigenin from bench to bedside.

#### 2.1.3. Other Flavonoids

There are many other flavonoids compounds that can induce autophagic cell death. Hydroxysafflor yellow A, obtained from safflower (*Carthamus tinctorius L*.), blocked the late-phase of autophagic flux by impairing lysosomal acidification and downregulating LAMP1 expression, therefore triggering the death of liver cancer cells [55]. Similarly, naringin-induced autophagic death in AGS gastric cancer cells also involves lysosomal damage [56]. Delicaflavone, a biflavonoid from *Selaginella doederleinii*, caused autophagic cell death in human non-small cell lung cancer via the Akt/mTOR/p70S6K signaling pathway, demonstrating positive anti-lung cancer effects with no observable side effects in a xenograft mouse model [57]. Flavokawain B from kava root killed human gastric carcinoma cells by inducing ROS-mediated autophagic cell death, involved in the suppression of HER-2/PI3K/AKT/mTOR signaling cascades [58]. Sinensetin, a polymethoxyflavone, induced the AMPK/mTOR pathway and subsequent autophagic cell death in HepG2 p53 wild type cells, and it induced apoptotic cell death in a p53-null cell line (Hep3B), suggesting sinensetin has great potential in the development of strategies for the treatment of apoptosis-resistant hepatocellular carcinoma [59]. In addition, the combination of flavonoids with chemotherapeutic agents has displayed a synergistic effect by regulating the levels of autophagy. This is evidenced by fisetin that sensitizes paclitaxel, causing a switch from cytoprotective autophagy to autophagic cell death in lung cancer [60]. Furthermore, liquiritin, a flavone derived from the medicine food homology plant liquorice, was also reported to enhance the anti-cancer effect of cisplatin by inducing both apoptosis and autophagy in cisplatin-resistant gastric cancer cells [61,62]. Taken together, these positive results provide a theoretical basis for flavonoids as therapeutic agents in the hope that they will overcome apoptosis-resistant tumors by causing autophagic cell death.

### 2.2. Improvement of Flavonoids Outcome by Suppressing Cytoprotective Autophagy

Faced with various therapeutic regimens, cancer cells endeavor to survive by activating cellular protection mechanisms. Cytoprotective autophagy, which is a key self-protection mechanism of cancer cells, is partially responsible for resistance and undermines the effectiveness of many anti-cancer treatments [23,63]. Accordingly, the strategy of inhibiting cytoprotective autophagy causing therapeutic resistance can improve the cytotoxic effects of the drug [64]. Indeed, the findings from both in vivo and in vitro studies have demonstrated that the anti-cancer activity of flavonoids can be impaired due to the activation of cytoprotective autophagy. Indeed, our own group has contributed strong evidence demonstrating that the inhibition of quercetin-induced cytoprotective autophagy could promote anti-cancer effects in gastric cancer [26].

Quercetin is a form of a polyphenolic flavonoid compound widely present in vegetables and fruits that has shown potential as a chemoprevention agent for various cancers due to the induction of apoptosis or lethal autophagy [65]. However, increasing studies are demonstrating that quercetin also activates cytoprotective autophagy, consequently compromising its anti-cancer effects. Indeed, a previous study from our group revealed quercetin stimulated autophagy by modulating HIF-1α-Akt-mTOR signaling pathways. Either autophagy inhibitors or siRNA-mediated autophagy blockage could enhance the pro-apoptosis effect of quercetin against gastric cancer cells [26]. Consistent with our study, quercetin was found to promote ER stress and concomitant protective autophagy that was associated with the activation of the p-STAT3/Bcl-2 axis, suggesting that autophagy scavengers could sensitize ovarian cancer to quercetin by ER stress-related apoptosis pathways [66]. In addition, Kim et al. [67] reported that quercetin inhibited the proliferation of malignant glioma cells by inducing cell cycle arrest and apoptosis that depended on the activation of JNK and upregulation of p53. Unexpectedly, quercetin also triggered protective autophagy evidenced by the augmentation of mitochondrial-mediated apoptosis after pre-treatment with the autophagy inhibitor chloroquine [67]. Quercetin-induced pro-survival autophagy was also identified in lymphoma and lymphoblastic leukemia [68,69]. Notably, clinical trials of quercetin are currently ongoing, mainly addressed to patients with prostate and kidney cancer (NCT01912820) (NCT02446795). The modulation of autophagy using quercetin may become a useful mechanism for improving cancer management.

In addition to quercetin, baicalein and apigenin as mentioned above can also induce protective autophagy in oral squamous cell carcinoma [70] or hepatocellular carcinoma, [71] respectively, suggesting that the application of these flavonoids based on autophagy inhibition would be a potential anti-cancer strategy for these tumors.

## 3. Flavonoids as Preventive or Adjuvant Agents

Dysregulation of autophagy, one of the emerging hallmarks of cancer [72] resulting in the reduced clearance of dysfunctional organelle and cytotoxic protein aggregates, accounts for oxidative stress-induced genomic instability accelerating oncogenesis [73]. The protective autophagy induced by flavonoids contributes to cellular homeostasis, thereby having great potential for preventing cancer occurrence (Figure 2). Similarly, flavonoids can also reduce side effects and protect healthy tissues from cytotoxic effects caused by chemotherapy or radiotherapy by inducing protective autophagy (Figure 2).

### 3.1. The Prevention of Cancer by Flavonoids-Induced Autophagy

Pre-malignant cells depend on autophagy to slow the onset of chronic inflammation that is a necessary and sufficient condition for tumorigenesis [74]. In fact, autophagy provides a robust safeguard for health tissues by degrading endogenous pro-inflammatory moieties and components of the inflammatory signaling networks including transmembrane protein 173 (TMEM173) and cyclic GMP–AMP synthase (cGAS) [75,76]. Put simply, restoration of autophagy activity or stimulation of efficient autophagic responses is a clever mechanism for intervention in the malignant transformation of healthy cells. In the context of this, one could consider using special diets of flavonoids to decrease the risk of cancer due to their anti-oxidant and anti-inflammation activities depending on autophagy.

#### 3.1.1. Delphinidin

Delphinidin is a type of anthocyanin primarily found in the maqui berry [77]. This subgroup of flavonoids have been identified as a potential candidate for health care applications in cardiovascular and neurodegenerative diseases [77]. Recently, an increasing number of studies have been directed at the use of delphinidin to target autophagy in cancer prevention. For example, levels of oxidized low-density lipoprotein (LDL) have been associated with the formation of haemangiomas [78,79]. A study by Jin et al. [80] reported that delphinidin-3-glucoside induced autophagy by activating the AMPK/SIRT1 axis, thereby protecting vascular endothelial cells from oxidized-LDL induced injury and implying a possibility of the use of delphinidin in haemangioma prevention. In addition, Lai et al. [81] demonstrated that high glucose could lead to apoptosis of pancreatic β cells whose dysregulation is closely linked with diabetes mellitus and associated cancers such as colorectal cancer [82,83]. Stimulation of autophagy by the AMPK signaling pathway using delphinidin treatment significantly diminished apoptosis in pancreatic β cells under high glucose conditions [83]. This study that explored the underlying mechanisms of the delphinidin-mediated protective role in pancreatic β cells could help to improve the prevention of diabetes-driven cancer. Recently, a study demonstrated that treatment with delphinidin prevented oxidative stress of human chondrocytes by regulating the Nrf2/NFκB networks and inducing intact autophagy [84]. Another study suggested that delphinidin had cytotoxic effects against human osteosarcoma cells by modulating autophagy [85]. Taken together, delphinidin may play a key role in tumor prophylaxis based on autophagy regulation.

#### 3.1.2. Silibinin

Silibinin is a flavonoid compound extracted from *Silybum marianum* L. Gaertn seeds and is well known for its hepatoprotective activities [86,87]. A recent study suggested silibinin-induced autophagy repressing hepatocyte injury caused by accumulation of acetaldehyde facilitated the progression of hepatocellular carcinoma through inflammatory responses [88,89,90]. Furthermore, silibinin was reported to have the potential ability of inhibiting non-alcoholic fatty liver cancer evidenced by the obvious amelioration of fructose-induced lipid accumulation by triggering autophagy [91]. Notably, silibinin also exhibited a cytoprotective effect by promoting autophagy and preventing apoptosis in pancreatic islet β-cells, a key therapeutic challenge involved in diabetes mellitus [92,93]. Accordingly, silibinin may be prophylactically beneficial for diabetes-associated cancer including pancreatic and colorectal cancer [82,94]. Abnormal endothelial cells that may result from insult contribute to the progression of cancer through excessive (lymph-)angiogenesis [95,96]. Silibinin was identified to have protective effects on human endothelial cells by activation of autophagy following high glucose induced damage [97], suggesting it as a candidate for cancer prevention. Excessive exposure to ultraviolet B is a major cause of skin damage including skin cancer. Silibinin is gaining acceptance as a chemo-preventive agent for UVB-irradiation-related skin cancer via modulation of autophagy [98] that is supported by multiple comprehensive studies. For example, Liu et al. [99] reported that silibinin induced protective autophagy by inhibition of the IGF-1R signaling pathway, therefore opposing UVB-irradiation-induced apoptosis. Despite the current absence of clinical trials using silibinin for cancer prevention, it would appear as a promising candidate agent based on solid preclinical data.

#### 3.1.3. Other Agents

Other flavonoids are also potential cancer-prevention agents. Epidemiological studies and clinical trials of isoflavone compounds have shown a range of beneficially preventative effects on breast cancer, endometrial cancer, and prostate cancer. (NCT00290758, NCT00099008, and NCT00617617) The preclinical studies clearly suggest that genistein as a typical example of isoflavone compounds could trigger autophagy pathways in a rat model. We therefore speculate that activation of the cytoprotective autophagy pathway may be a reasonable explanation for the cancer prevention potential of isoflavones [100,101]. Similar to silibinin, isoorientin was recently reported to demonstrate inhibition of UVB-induced damage by decreasing the ROS accumulation and activation of JNK signaling, increasing autophagy flux [102]. Li et al. [103] revealed that malvidin-3-O-arabinoside, a form of anthocyanin extracted from blueberries, displayed a significant efficacy against oxidative damage triggered by ethyl carbamate (EC), which is listed as a universal carcinogen [104]. Mechanically, malvidin-3-O-arabinoside was shown to activate autophagy flux by enhancing phosphorylation of AMPK and expression of LAMP-1, thus contributing to the remission of EC-caused oxidative damage to the intestinal epithelial cells [105]. In addition, pelargonidin-3-O-glucoside derived from wild raspberries induced TFEB-mediated autophagy and modulated gut microbiota in an animal model, suggesting a potential intervention strategy for preventing the cancer progression of patients with a disturbance of their intestinal flora [105].

### 3.2. Use of Flavonoids to Improve Side Effects Caused by Treatment

Chemo and radiotherapy are still the most commonly employed treatments for advanced tumors regardless of frequent acute and chronic adverse effects on non-malignant tissues [106]. Autophagic responses have been revealed as a key defense for maintaining normal cellular homeostasis and mediating oncosupressive effects [107,108]. Thus, a combinatorial strategy with natural compounds such as flavonoids that stimulate the reactivation of autophagy during chemo or radiotherapy potentially can improve treatment side effects and even prevent cancer development. In this content, we will discuss the protective properties of flavonoids against toxicity during cancer therapy by inducing autophagy.

#### 3.2.1. Epigallocatechin-3-Gallate (EGCG)

Epigallocatechin-3-gallate (EGCG) is a main catechin in green tea and has been revealed to have a promising protective role against side effects of chemo and radiotherapy [109]. A study has demonstrated that EGCG partially alleviated cisplatin-elicited nephrotoxicity by regulating enzymes of carbohydrate metabolism, the brush border membrane, and enhancing antioxidant signaling and inorganic phosphate transport [110]. This EGCG-mediated process may also be associated with autophagy activation in the kidney [111]. In agreement, Kazim et al. [112] also reported EGCG had a preventive ability against cisplatin-induced nephrotoxicity by relieving oxidative damage by triggering Nrf2/HO-1 signaling. Furthermore, EGCG attenuated doxorubicin-induced cardiac injury in neonatal rats by modulation of redox signaling including the activation of catalase, manganese superoxide dismutase (MnSOD), and glutathione peroxidase [113]. This implies that EGCG likely resulted in the induction of autophagy as it is usually accompanied by Nrf2-mediated redox signaling activation to combat oxidative damage and maintain homeostasis [114]. EGCG has also been shown to contribute to cardioprotective effects in doxorubicin-treated tumor-bearing mice [115]. Due to the well-known crosstalk between redox signaling, cellular calcium homeostasis, and autophagy [116], autophagic responses are potentially involved in the EGCG-mediated reduction of myocardial calcium overload and increase of MnSOD expression [117]. For radiation injury, EGCG protected submandibular glands from radiation-induced apoptosis potentially through an autophagic process [118,119].

#### 3.2.2. Other Flavonoids

The natural flavone luteolin, a common flavonoid present in many types of plants including fruits, vegetables, and medicinal herbs [120] was recently reported to exhibit anti-cardiotoxicity properties in doxorubicin-treated cardiomyocytes. Mechanistically, luteolin reversed cardiotoxicity caused by doxorubicin by inducing mitophagy and causing increased phosphorylation of Drp1 at Ser616 and transcription of TFEB [121]. Consistent with this, He et al. [122] indicated that curcumin extracted from *Curcuma longa* L. attenuated the side effects of doxorubicin by inhibiting mitochondrial dysfunction and reducing oxidative stress with a potential relationship to mitophagy. Furthermore, aspalathin, an active ingredient of *Aspalathus linearis*, also displayed a similar ability against doxorubicin-induced cardiotoxicity. Aspalathin-triggered autophagy contributed to this process by the activation of AMPK and Foxo1 [123]. Pre-treatment with an active flavonoid from licorice, isoliquiritigenin, was revealed to compromise cisplatin-induced nephrotoxicity by activating unfolded protein response (UPR) during ER stress [124]. We speculate that isoliquiritigenin-mediated UPR reduces oxidative stress by induction of autophagy [116,125]. Additionally, the use of natural products such as rutin, a citrus flavonoid found in plants, with chemotherapeutic drugs can help to reduce the toxicity associated with conventional drugs and enhance their therapeutic efficiency [28,126]. Ma et al. also reported the suppression of autophagy and apoptosis, demonstrating administration of rutin could attenuate doxorubicin-induced cardiotoxicity [127]. These findings enhance our knowledge for developing new drugs and strategies for combating the potentially devastating side effects induced by conventional drugs.

## 4. Perspective and Conclusions

Alongside extensive studies on drug discovery, the evidence presented here demonstrates that drug repurposing of flavonoids could provide a novel approach to cancer prevention and therapy, and is also a potential strategy for further studies on the development of new cancer preventive agents or antineoplastic drugs [128]. It is worth noting that flavonoids promoted different cellular effects by modulating multifunctional autophagy, suggesting their potential use in different stages of cancer. For example, flavonoids that induce autophagic cell death by either excessive autophagy or autophagy flux blockage may be more effective in cancer therapy, especially for apoptotic resistance cancer. Importantly, combinational strategies should be considered if they suppress the growth of cancer cells and concurrently induce protective autophagy. In contrast, flavonoids that exhibit the properties of reactivation of autophagy may be more suitable for cancer prevention including inhibition of cancer initiation and alleviation of chemo and radiotherapy-induced side effects or secondary cancer (Figure 2). In this review, focusing on the regulation of autophagy, we have overviewed a number of flavonoids that can act at different stages of cancer development. Based on compounds such as quercetin that displays potential for both cancer prevention and therapy by controlling two aspects of autophagy, we suggest that flavonoids should be further investigated to reveal their detailed mechanisms of action that would help facilitate optimum therapeutic administration.

In addition to a lack of understanding of the intricate mechanisms of action of flavonoids, the low bioavailability and non-specific selectivity generally hinders their pharmacological applications in clinical settings. It has been suggested that these issues could be overcome by designing nano-engineering systems to improve the targeted delivery of flavonoids, resulting in better efficacy and improved pharmacokinetic properties [129]. Studies have shown that some flavonoid nanoparticles such as quercetin, catechins, and apigenin offer improved water solubility and tumor site-specific targeting, thus further enhancing their effectiveness for cancer prevention and treatment [130] (Figure 3). Notably, a clinical trial of nano-flavonoids using nano-luteolin has been initiated to assess whether it exerts an inhibitory effect on tongue squamous cell carcinoma by inducing apoptosis (NCT03288298). This heralds a promising future regarding the use of nanoparticles for delivering flavonoids for cancer prevention and therapy.

In conclusion, we hope that this review will be of use to researchers and clinicians by offering a more profound understanding of the application of flavonoids for treating cancer patients and susceptible populations based on modulating the dual role of autophagy.

## Figures and Tables

**Figure 1 antioxidants-10-01138-f001:**
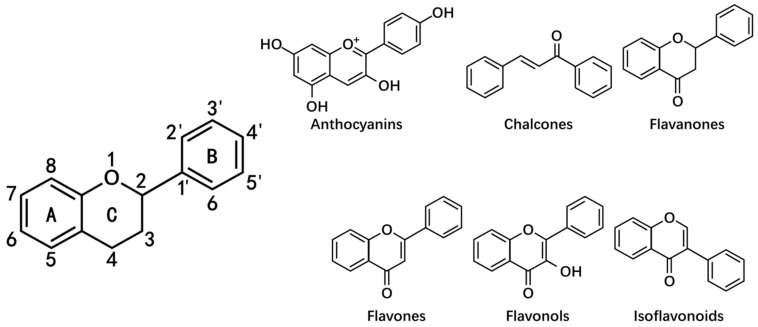
The basic structure of flavonoids and their six main subgroups. This figure was created using BioRender.

**Figure 2 antioxidants-10-01138-f002:**
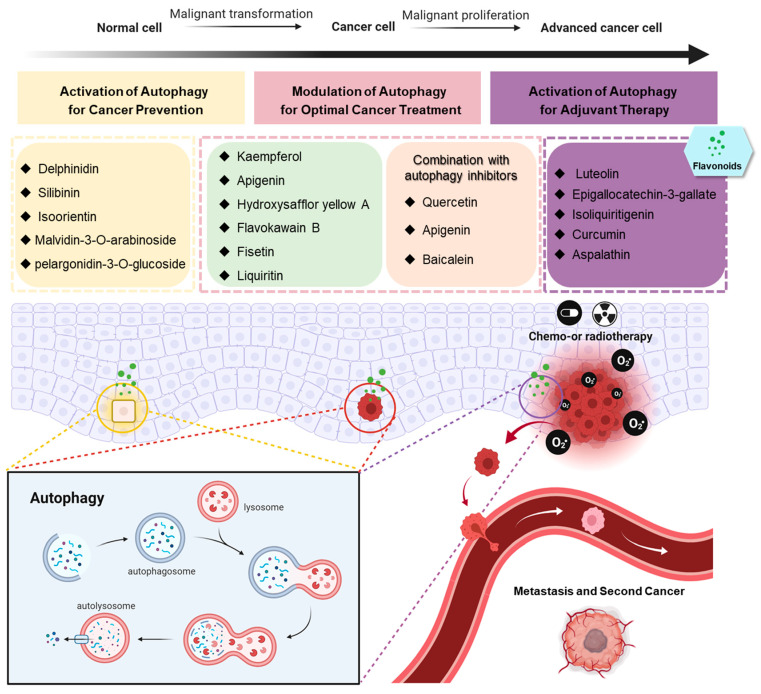
Flavonoids play a different role in various stages of cancer, attributing to the modulation of multifunctional autophagy. This figure was created using BioRender.

**Figure 3 antioxidants-10-01138-f003:**
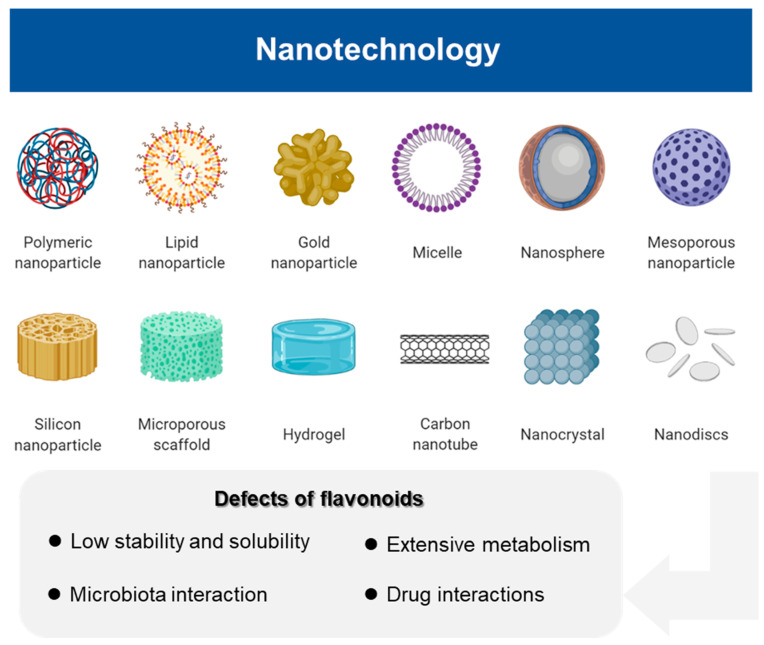
Nanotechnology partially overcomes the defects of flavonoids such as low stability and solubility, extensive metabolism, microbiota interaction, and drug interactions, and further contributes to their effectiveness and drug targeting. This figure was created using BioRender.

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
