# Peer review of "The Multifaceted Role of Flavonoids in Cancer Therapy: Leveraging Autophagy with a Double-Edged Sword"

_antioxidants, 2021, doi:10.3390/antiox10071138_

Round 1

Reviewer 1 Report

The article entitled "The multifaceted role of flavonoids in cancer therapy: leveraging autophagy with a double-edged sword" reviews the information about  autophagic mechanisms of flavonoids in clinical application. 

In general terms, the article is organized, straight-forward and I cannot find big discrepancies. However, some suggestions will be given in the following lines, to improve the article.

Line 14: I recommend correcting "natural products" into "molecules", "chemicals".

The number of keywords could be also improved to make the article more recognizable. 

Line 29: authors should once more rewrite this phrase: "drug discovery of drugs".

Before paragraph 2.1.1 a short sentence for introduction to this part should be added. It will make he text easy to follow and be more readible.

Latin names of plants should be written in italis e.g. Line 147.

Please check the references in the text e.g. Line 314: it should be "He et al. [117]..." The same in Line 133, 259, 297.

To resume, authors have prepared an interesting manuscript. The manuscript is clear. The impressive number of 123 references was cited and the literature is up-to-date. Only a few issues should be improved before final publication. Therefore, I am suggesting MINOR REVISION.

Reviewer 2 Report

This review article, focusing on the underlying autophagic mechanisms of flavonoids,  proposes potential clinical applications of flavonoids in cancer therapy. It is of interest and it is suitable for publication, but after a major revision.

1) I suggest to insert in Intraduction section a part in which Authors report the years covered by their reseach, the used keywords, the used databasis, the inclusion and exclusion criteria.

2) A paragraph (also brief!!!) that describes the chemical characteristics of flavonoids must be inserted.

3) Paragraph "2.1.3. Other flavonoids" must be improved.

4) Paragraph "3.1.3. Other agents" must be improved.

5) Paragraph "3.2.2. Other flavones" must be improved.

6) Figure 3 should not be placed at the end of the conclusions.

Reviewer 3 Report

Authors present a current overview on the use of flavonoids in  cancer prevention and cancer therapy and its possible mechanisms of action. Even when flavonoids and other botanicals are not yet routinely used in clinical cancer therapy, they represent promising candidates and alternatives to chemo- and radiotherapy. 

The review is comprehensive and thoroughly done, but the manuscript needs some edotorial and language improvements.

(1) All species names mantioned in the texz as well as in the References have to be written in italics.

(2) The references have to be presented in a uniform style (all paper titles in lowercase letters, etc.)

  • line 19: is becoming a trend...
  • line 29: for discovery of drugs...
  • line 52: better say isoflavonoids, because there is not only one
  • line 106: Kampheria galanga L. is the correct name
  • line 130 and others: et al. is an acronym and has to be written with a dot
  • line 268 and others: use lowercase letters for all chemicals, if not at the beginning of a sentence
  • line 314 and others: give corect names for plant species, e.g. Curcuma longa L.

Round 2

Reviewer 1 Report

The Authors improved the manuscript and included all the suggestions. In this form, it could be accepted to be published.

Reviewer 2 Report

This work is now suitable for publication.

Reviewer 3 Report

The manuscript has been well improved.